# Associations between Deformation of the Thoracolumbar Fascia and Activation of the Erector Spinae and Multifidus Muscle in Patients with Acute Low Back Pain and Healthy Controls: A Matched Pair Case-Control Study

**DOI:** 10.3390/life12111735

**Published:** 2022-10-28

**Authors:** Andreas Brandl, Christoph Egner, Rüdiger Reer, Tobias Schmidt, Robert Schleip

**Affiliations:** 1Department of Sports Medicine, Institute for Human Movement Science, Faculty for Psychology and Human Movement Science, University of Hamburg, 20148 Hamburg, Germany; 2Department for Medical Professions, Diploma Hochschule, 37242 Bad Sooden-Allendorf, Germany; 3Osteopathic Research Institute, Osteopathie Schule Deutschland, 22297 Hamburg, Germany; 4Institute of Interdisciplinary Exercise Science and Sports Medicine, MSH Medical School Hamburg, 20457 Hamburg, Germany; 5Conservative and Rehabilitative Orthopedics, Department of Sport and Health Sciences, Technical University of Munich, 80333 Munich, Germany

**Keywords:** EMG, ultrasound imaging, low back pain, thoracolumbar fascia, erector spinae muscle, multifidus muscle, muscle spindle

## Abstract

Background: The thoracolumbar fascia (TLF) is thought to play a role in the development of LBP, but it is not yet clear which factor of TLF changes is a cause and which is an effect. Therefore, some studies used the cross-correlation function (CCR) to reveal time-dependent relationships between biomechanical and neuromotor factors. Methods: Ten patients with acute low back pain (aLBP) were matched to healthy controls. Simultaneous recording of surface electromyography (sEMG) of the erector spinae and multifidus muscle (ESM) and dynamic ultrasound (US) images of TLF deformation were performed during trunk extension. CCR functions and Granger causality (GC) were used to describe the relationship between the two measures. Results: CCR time lags were significant higher in the aLBP group (*p* = 0.04). GC showed a direct effect of TLF deformation on ESM activation only in the aLBP group (*p* < 0.03). Conclusions: The results suggest that in aLBP, ESM activity is significantly affected by TLF, whereas this relationship is completely random in healthy subjects studied with CCR and GC comparisons of dynamic US imaging and sEMG data signals. Fascia-related disturbances in neuromotor control, particularly due to altered muscle spindle functions, are suspected as a possible mechanism behind this.

## 1. Introduction

Nonspecific low back pain (LBP) is one of the main causes for visiting an orthopaedist or manual therapist [1,2]. In addition, LBP goes hand in hand with significant disability, work loss, and early retirement, making it a critical cost and resource factor in health care systems [3,4]. The prevalence of nonspecific LBP has been reported to range from 30% to 70% in the age group between 18 and 74 years [5]. The point prevalence is 30% worldwide [6], and the trend has been increasing over the last 20 years [7]. Considering the growing industrialization in developing countries and the rising number of elderly people in the society [8], which are major risk factors for the development of LBP [4], LBP is also a serious health problem in the coming decades.

Many clinicians and scientists have been working over the past decades to find associations or risk factors for predicting the development of LBP [4,6,7,8]. Since there is no clear cause, and even disc disease or radiculopathy cannot be placed in a linear causal relationship with LBP, a multifactorial aetiology has become a widely accepted theory [2,5,9]. However, there are some risk factors with high odds ratios (OR) that could help to partially predict the development of LBP. Parreira et al. [4], in an umbrella review that included 15 systematic reviews of moderate to high quality, identified factors such as previous LBP (OR 1.2−4.5), older age (OR 1.3−5.9), obesity (OR 1.1−2.4), or physical exertion (OR 1.2−6.4). Phases of acute LBP (aLBP) in the history is one of the main risk factors for the development of chronic LBP, along with psychosocial factors [4,10]. 

For biomechanical causes, far fewer demographics are reported in the literature. Some authors suggested microinjuries of the paraspinal connective tissue and the thoracolumbar fascia (TLF) as a possible cause of LBP [11,12,13]. Recent studies emphasize the nociceptive role of the high proportion of nonmyelinated terminal nerves in the TLF, which could lead to long-lasting sensitization of dorsal horn neurons in response to microinjuries [9,11,14]. Adhesions between the TLF layers and the epimysium of the erector spinae and multifidus muscle (ESM) are considered to be an effect of LBP pain rather than its cause [11,15]. The limitation of daily movements of patients suffering from LBP is thereby seen as a possible explanation [15]. On the other hand, some authors emphasized the association of TLF adhesions or flaccidity with diseases of adjacent lumbar segments and altered motion patterns, but it remains unclear whether the TLF is the cause or vice versa [11,15].

There is significant alteration in muscle activation patterns in patients suffering from LBP [16,17]. Proprioception is also affected, particularly in the TLF [18]. In addition, the central nervous system has shown correlative changes associated with LBP [19]. Furthermore, electromyographic (EMG) studies revealed significantly different spatial distributions of erector spinae muscle activity between patients with LBP and healthy controls [20,21]. Wilke et al. [15] hypothesized that sensitization of nociceptors in the TLF, less refined proprioceptive afferent input, and stimulation of other tissues innervated by the same spinal segment influence each other. This leads to changes in muscle activation and movement patterns, which in turn impacts the aforementioned factors. However, it remains to be elucidated which factor is a cause and which an effect, or whether there is even a linear causal relationship between the individual factors.

Previous work has tried to find such correlations with varying results [22,23,24,25,26]. The behaviour of the myofascial tissue has been shown to be very individual and does not seem to be the same even in two people. Some studies examined the direct correlation between ultrasound (US) and EMG data [24,25]. Other authors attempted to compare joint range of motion and US data [27]. Some authors have taken into account that the relationships between different variables often occur with a time lag and have therefore used cross-correlation (CCR) of simultaneous time series instead of single variables for the analysis [26,28]. In psychology and neuroscience, such studies are common, especially when examining perception and performance [29]. Wren et al. [30] recommend the use of CCR analysis particularly for the examination of dynamic EMG signals, highlighting that it is more objective than the usually used qualitative visual assessments of muscle timing.

### Aims

Based on previous work, the authors hypothesized that adhesions of the TLF to the epimysium of the ESM could alter its muscle spindles (MS) and thereby affect motor control [31]. Theoretically, Stecco et al. [32] suggested the MS could be blocked because of the tight connection between the muscle and fascia if the ability of the tissue to slide against each other is reduced. To reveal such relationships, TLF deformation during an ESM task was measured using dynamic ultrasound (US) and converted into a two-dimensional signal (time, distance) [8]. Simultaneously, ESM activity was recorded and analysed for similarity to the US signal using CCR function. 

In addition, it was assumed that TLF deformation at an early stage of LBP might affect ESM MS more than ESM activation affects TLF. Therefore, the direction of the effect of whether the EMG signal predicts the US signal (ESM activation causes TLF deformation) or vice versa (TLF deformation causes ESM activation) was tested using GC [29]. 

If these hypotheses can be proven, measurable relationships between the fascial system and muscles could be of great importance in clinical practice to offer early diagnosis and treatment to patients with acute low back pain (aLBP), helping to avoid chronification.

## 2. Materials and Methods

### 2.1. Study Design Overview

The study was an individual, one-to-one matched case-control validation study according to the STROBE Statement [33]. The study protocol was prospectively registered with the German Clinical Trials Register (DRKS00027074) on 5 November 2021. The study was reviewed and approved by the ethical committee of the Diploma Hochschule, Germany (Nr.1014/2021), was carried out in accordance with the declaration of Helsinki and obtained informed consent from the participants [34]. 

### 2.2. Setting and Participants

The study was conducted in an osteopathic practice in a medium-sized city in southern Germany. The number of participants for the dependent groups was calculated based on experimental examinations on 3 patients with aLBP and 3 healthy subjects (Cohen’s d = 0.7, α err = 0.05, 1–β err = 0.9) and set at 10 per group [35]. The acquisition for the aLBP group was carried out via direct contact, a notice board and the distribution of information material in the practice. The control group was matched with cases according to age (±5 years), sex and BMI (3 classes: “normal”, BMI between 18.5–24.9; “pre-obesity”, BMI between 25.0–29.9; “grade 1 obesity”, 30.0–34.9).

#### 2.2.1. Inclusion Criteria for the aLBP Group

Inclusion criteria were: (a) acute lumbar back pain (aLBP) as defined by the European guidelines for the management of acute low back pain [27]; (b) a minimum score of 10 on the Oswestry disability questionnaire in the German version (ODQ-D) [28]; (c) a minimum score of 3 on the visual analogue scale (VAS) for assessment of pain intensity [29]; (d) less than 6 weeks pain duration; (e) a BMI between 18.5 and 34.9.

#### 2.2.2. Exclusion Criteria

Exclusion criteria were: (a) generally valid contraindications to physiotherapeutic and osteopathic treatments of the lumbar spine and pelvis; (b) rheumatic diseases; (c) taking medication that affects blood coagulation or drug treatment of endocrine diseases; (d) taking muscle relaxants; (e) skin changes (e.g., neurodermatitis, psoriasis, urticaria, decubitus ulcers); (f) surgery or other scars in the lumbar region between Th12 and S1; (g) acute trauma; (h) neurologic or psychiatric disorders; (i) subjects under 18 or over 60 years of age; (j) prone position for 15 min is not painless for the subjects. Additional exclusion criteria for the control group were either (k) the presence of current LBP or (l) a history of LBP (no LBP episodes in the past 5 years; no history of physician visits due to LBP).

### 2.3. Procedure

The volunteers were first screened for eligibility by the investigator. The aLBP group completed the ODQ-D [36] and determined their current pain perception on the VAS. Prior to the measurements, the subjects received information on how to perform a defined trunk extension task (TET). The investigator demonstrated a complete cycle of the TET (Figure 1). Participants first performed 45 degrees of thoracolumbar flexion measured with a digital goniometer at the level of Th10 [37].

Study participants sat on the treatment table with their feet touching the floor [38]. Subjects were then instructed to place their hands lightly on their thighs and keep their elbows close to their bodies. They then performed a slow 45-degree flexion of the upper body without moving the arms. In doing so, the investigator stopped the movement when the goniometer indicated the appropriate value. They then extended the trunk to the neutral position (0 degrees) over a period of 8 s. A computer-generated time announcement provided temporal guidance.

The examination and treatment were carried out by the owner of the individual practice for osteopathy (AB).

### 2.4. Outcomes

#### 2.4.1. Ultrasound Measurement of the Deformation of the Thoracolumbar Fascia

Measurement of TLF deformation using the latissimus dorsi muscle (LD) as an anatomical landmark for dynamic ultrasound (US) imaging against an artificial reference was previously described by Wong et al. [39]. Jhu et al. [40] tested this ultrasound approach to measure myofascial length changes and determined an ICC = 0.98 for lateral sliding of the muscle–fascia junction.

First, the processi transversi of L1 were sonographically recorded (Mindray DP2200, linear transducer 75L38HB, 5–10 MHz, sampling rate 7.5 MHz) and marked with a 10 cm line drawn with a pen. A lot was used to select the right or left side of the body for examination. The transducer was then moved laterally along the line from the L1 spinous process in the sagittal section until the junction of the LD muscle with the TLF was visible. While this remained centrally aligned on the transducer, it was rotated latero-caudally until the fibres of the LD were aligned in parallel. The junction between muscle and fascia was then positioned 1.5 cm medial to the centre of the image section. An artificial shadow was created in the US image using a 2 mm wide plastic adhesive tape placed on the skin 1.5 cm lateral to the centre of the image section, which served as a reference for the subsequent measurement [41]. This was necessary to compensate for unintended movement between the skin and the transducer during TET [39,40].

The deformation of the *TLF* was defined by the distance between the intersection of the artificial reference and the underside of the posterior layer of the TLF (*X*2|*Y*2) (Figure 2, green dot) and the muscle–fascia junction of the LD and the *TLF* (*X*1|*Y*1) (Figure 2, orange dot), as described by Willard et al. [42] and Stecco [43]. X represents the coordinate point on the x-axis with respect to the left edge of the image, and *Y* represents the coordinate point on the y-axis with respect to the bottom edge of the image. The two-dimensional coordinate distances were calculated according to the following Equation (1):(1)$$\Delta TLF=\sqrt{{\left(X2-X1\right)}^{2}+{\left(Y2-Y1\right)}^{2}}$$

The transducer position thus determined was marked by an outline with a pencil and represented the region of interest (ROI) for the examination (Figure 3). During TET, a real-time video recording of the B-mode US measurement was made with a resolution of 852 × 340 pixels and a frame rate of 15 frames per second. The deformation of the TLF was calculated by evaluating the ROI of the frames using Tracker Video Analysis and Modeling Tool version 6.0.5 (© 2022 Douglas Brown, Wolfgang Christian, Robert M. Hanson; GNU General Public License, Version 3). The technical accuracy of the measurement, which was determined by the maximum image resolution, is ±118 µm. The generated dataset was saved as a table in OpenDocument format. For the total deformation of the TLF, the coordinate distances were measured once at the start of the TET in the flexed trunk position (first video frame) and once in the stretched trunk position (last video frame), and the difference was calculated. An example of a measurement with the Tracker Video Analysis Tool can be found in Appendix A.

#### 2.4.2. Surface Electromyography of the Erector Spinae and Multifidus Muscle

The surface EMG (sEMG) electrode pairs were placed, according to the SENIAM protocol [44], at the level of L1 (longissimus muscle) as well as 2 cm cranially and at the level of L5 (multifidus muscle) as well as 2 cm caudally and 2 cm paravertebrally of the spinous process, on the side of the ESM determined by means of a lot for bipolar conduction (Figure 3). Hofste et al. [45] compared needle EMG and sEMG in a correlation study. According to Cohen [46], they determined a high correlation between the needle EMG of the longissimus muscle (r = 0.9) and the multifidus muscle (r = 0.8) for a biofeedback task. 

The reference electrode was positioned centrally at the level of the 3rd sacral vertebra. All skin contact sites were prepared for maximum conductivity using an abrasive paste followed by alcohol cleaning. Round self-adhesive 15 mm Ag/AgCl wet gel electrodes were used. 

The sEMG signal was acquired by an EMG amplifier (Biosignalplux, Senpro-EMG1, 25–500 Hz, sampling rate 3 kHZ, 16-bit A/D conversion) with a common-mode rejection ratio (CMRR) of 110 dB and an input impedance of 110 GΩ during TET and transmitted via Bluetooth to a PC, where it was recorded and stored in OpenDocument format, using the software Opensignals version 2022-05-16 (Biosignalplux, Lisboa, Portugal).

#### 2.4.3. Data Synchronization

The synchronization of the US recording and the sEMG signals was performed via a 1.5-volt DC signal, which was triggered by the investigator with a button at the beginning of the TET. This signal was detected by one channel of the sEMG device and was visible as a peak in the sEMG signal. The signal was also stored as an audio signal parallel to the US video recording and was visible there as a voltage peak in the evaluation software. The signals could be synchronized in this way with a minimum tolerance of 10 ms.

#### 2.4.4. Cross Correlation Analysis of the Measurement Series

The raw sEMG data were processed with the R package “bioSignalEMG” and scaled in µV. The root mean square (RMS) was calculated from the signal, normalized to the maximum amplitude and transformed using Principal Components Analysis (PCA) whitening. This was to prevent autocorrelation of the sEMG signal from affecting the cross-correlation analysis.

A reduction of the sEMG data, sampled to the timestamp of the US video image, was performed so that 15 US images per second could be assigned to 15 sEMG datasets in time synchrony. The largest correlation coefficient (rCCF) of the CCR function indicated the degree to which the sEMG and US signals were temporally associated with each other. Here, the two signals were shifted against each other by a certain time lag (τLag) until they were maximally congruent (Figure 4). Since the interpretation of CCR of two time series alone can be problematic, a test for GC was performed for each measurement [29]. Here, the null hypothesis that the sEMG signal was not a cause of the US signal was tested using a significance level of *p* = 0.05.

### 2.5. Statistics

Student’s *t*-tests (for normally distributed data) and Mann–Whitney U tests (for non-normally distributed data) were used to compare groups with respect to baseline characteristics.

The standard deviation (SD), mean and 95% confidence interval (95% CI) were determined for all outcome parameters. There were no outliers in the data. The outcome variables were normally distributed as determined by the Kolmogorov–Smirnov test (*p* > 0.05). 

Student’s *t*-test for dependent samples was performed to detect group differences between TLF deformation, τLag and GC; between TLF deformation and sEMG at L1 level, τLag and GC; and between TLF deformation and sEMG at L5 level. Multiple tested *p* values were adjusted according to Bonferroni–Holm, and effect sizes were calculated according to Cohen’s d. Following Cohen [42], these values were interpreted as “weak” (>0.09, <0.30), “moderate” (>0.29, <0.50) and “strong” (>=0.50) [46]. Estimation plots were used to additionally visually control for group differences [47].

Libreoffice Calc version 6.4.7.2 (Mozilla Public License v2.0) was used for descriptive statistics. Inferential statistics were performed using R software, version 3.4.1 (R Foundation for Statistical Computing, Vienna, Austria).

## 3. Results

Twenty subjects were successfully matched according to age, sex and BMI classes. No significant difference in baseline data were found between the groups (Table 1).

Table 2 shows a significant difference in TLF deformation between groups in row 1 (see also Figure 5). In rows 2 and 3 (L1/L5 time lag) we observe that there were significant differences in the temporal association of TLF deformation and ESM sEMG activity at the level of L1 and L5 between groups. In the aLBP group, there was the greatest correlation of the time series in the positive range (TLF deformation was followed by sEMG activity), whereas in the control group these occurred mostly in the negative range (sEMG activity was followed by TLF deformation) (see also Figure 6). Rows 4 and 5 (GC) show a directional effect, i.e., here, the time series of the TLF deformation predicted with high probability the time series of the sEMG activity of the ESM.

Table 3 shows the probability to which the time series of TLF deformation is suitable to predict the time series of ESM sEMG activity or vice versa. The mean values of the groups are shown in percentages (see also Figure 7, where a further differentiation by matched pairs was made).

## 4. Discussion

The present matched case-control validation study is novel in that it examines the relationship between TLF deformation (measured with US) and paraspinal muscle activity (measured with sEMG of ESM) during a clinically relevant task in subjects with and without aLBP. In addition, this is the first study to apply CCR functions and GC to lumbar myofascial tissue, which are commonly used in signal processing analysis in psychophysiology and neuroscience [29]. Whittaker et al. [26] used a similar methodology to compare abdominal muscle thickness and EMG activity. To the authors’ knowledge, this was the first study to use CCR functions and GC to reveal the interactions between TLF and ESM in pathological and healthy subjects. This approach allows us to compare the similarity of two signals (waves) in terms of time and shape as a function of the time delay that may occur between them.

**Figure 7 life-12-01735-f007:**
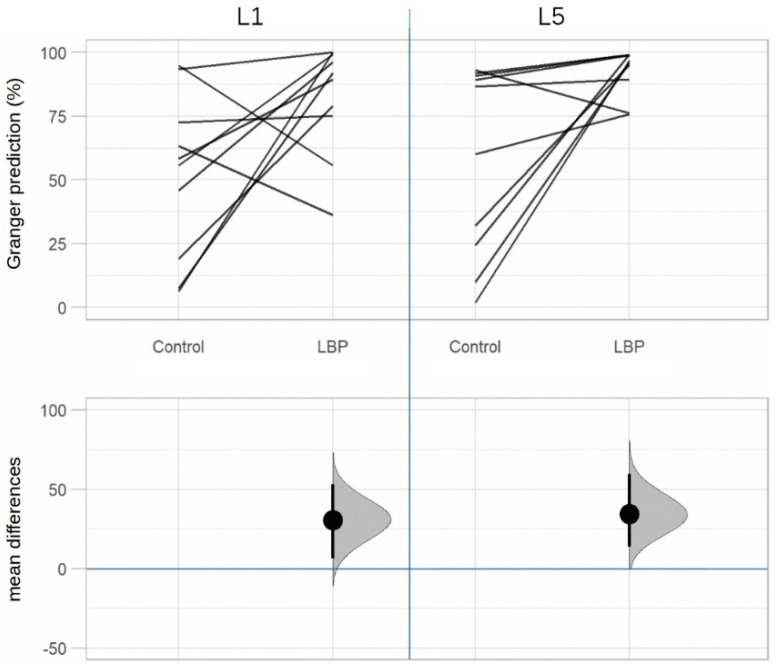
Estimation plot of the Granger causality: TLF deformation causes sEMG activity. TLF, thoracolumbar fascia; sEMG, surface electromyography; LBP, acute low back pain group; Control, control group; both significant at level *p* < 0.05.

The results of both CCR and GC showed significant differences in the relationship between the TLF and the ESM during a TET in the aLBP group (all *p* < 0.05). Here, TLF deformation was clearly followed by sEMG activity in the ESM. GC revealed an 82.3% probability that fascia influenced muscle activity at L1 level (*p* = 0.03) and 92.5% at L5 level (*p* = 0.02). However, this phenomenon occurred only in subjects with aLBP but not in the control group, in which the relationship between fascia and muscle was found to be completely random (all *p* > 0.69); GC revealed a 50:50 probability that one variable predicted the other. There was a significant difference in total TLF deformation between aLBP patients and healthy controls (*p* = 0.01). Using the same US measurement method, Wong et al. [39] observed a difference in TLF deformation of 3.8 mm after myofascial release treatment and attributed these changes to the decrease in stiffness due to treatment. In the healthy control subjects, the TLF was 27.5% more deformable during TET than in the aLBP cases. According to the results of Wong et al., this could be due to a higher stiffness of the TLF tissue. 

During TET, we observed in some aLPB participants with putative adhesions that the ESM did not show continuous smooth contraction but multiple spasmodic twitches. Interestingly, tearing of the putative adhesion appeared to be associated with severe lumbago pain (Appendix A). The muscle spindles of the ESM are completely in continuity with the intramuscular connective tissue, especially with the perimysium and, via fascial continuity, also with the epimysium [48,49,50]. Adhesions between the TLF and the epimysium of the ESM could transmit mechanical forces to the muscle spindles during a dynamic task, triggering a stretch reflex, which we may have observed in some participants with pathological changes in the TLF (Figure 8 and Figure 9). If the adhesions between the TLF and epimysium persist for a prolonged period of time, the muscle spindles associated with this area are likely to be stretched and thereby activated. Clinicians often describe unilateral hypertrophy of the ESM in LBP patients at the level of the lower thoracic spine to the L3 or L5 vertebrae. Interestingly, this is the area with the highest density of muscle spindles in the posterior vertebral muscles [51], so the corresponding muscle fibres could be constantly contracted by fascially triggered muscle spindle activation [32]. This is consistent with our clinical observations that such TLF adhesions also preferentially occur unilaterally. 

Even a load threshold of 30 mN is capable of activating a muscle spindle. Therefore, the elasticity of the peri–epimysium–fascial complex is a critical factor in maintaining proper muscle spindle function [32,52]. The signals from the spinal cord to the muscle are often irregular and not uniform. The intensity initially increases for a few milliseconds, then decreases, changes to a different intensity level, and so on. The muscle spindle smooths this signal by mediating the spindle reflex to average the signal [52]. Moreover, because of α–γ-coactivation during muscle contraction, the muscle spindle must shorten accordingly in response to this efferent signal [32]. Adhesions of the TLF to the epimysium of the ESM reduce the elasticity and adaptability of the intramuscular connective tissue to which the spindles are connected and probably thereby alter their proper physiological function.

Conversely, not only can the muscle spindle be stretched or blocked by the surrounding connective tissue, but the intrafusal fibres themselves can also transmit a force via endo-, peri- and epimysium due to fascial continuity. Fede et al. [49] suggest that the nerve network in the epimysial fascia can unify all the tension emanating from the muscle spindles and transmit decoded input about the state of muscle contraction via the central nervous system. Therefore, in this context, the epimysial fascia acts as a kind of coordinator of the actions of the different muscle motor units. Pathological alterations of the TLF associated with decreased shear capacity relative to the epimysium would obviously interfere with such a coordinating function by mechanically blocking the receptors [11].

### Limitations

First, in this study, the junction between LD and TLF was used as a landmark for dynamic US imaging during TET. Participants were instructed to avoid upper limb movements. However, it is possible that LD activity could still have influenced the measurement. Wong et al. [39], who described this methodology, examined three trials for how TLF was affected by LD activation and found no significant differences between trials. However, future studies should also test LD activity in relation to EMG silence to avoid potential bias.

Second, the sample size was relatively small. In preparation for this study, an experimental examination was performed on six subjects. The derivation and analysis of the results, especially the dynamic US images, took 60 min for each participant and was therefore very resource-intensive. Therefore, we carefully calculated the necessary sample size with the experimentally determined effect size and increased the statistical power with a matched-pairs design. Our results showed strong effect sizes and a high statistical power for CCR (d = 0.85, 1–β err = 0.98 at L1 level; d = 0.81, 1–β err = 0.97 at L5 level) and GC (d = 0.69, 1–β err = 0.91 for TLF/sEMG relationships at L1 level; d = 0.85, 1–β err = 0.98 at L5 level), indicating that the sample size chosen was large enough to reveal significant differences between aLBP participants and healthy controls.

Mechanical forces to the muscle spindles during a dynamic task, triggering a stretch reflex.Muscle spindles are blocked by adhesions, and their function is disturbed, which is why they are unable to smooth muscle contraction.An epimysium incapable of shearing against the TLF cannot provide a proprioceptive function to unify muscle spindle tensions.

Some authors urged caution in the use of sEMG, especially in the multifidus muscle [45,53]. Hofste et al. [45] showed a high correlation between needle EMG and EMS sEMG. However, they also documented a high rate of crosstalk during muscle coactivation tasks (in which they did not follow the SENIAM protocol and intentionally provoked coactivation). Bandpei et al. [54] instead highlighted the use of sEMG in assessing paraspinal muscle fatigue and certified sEMG as reliable. However, future work is needed to further demonstrate the accuracy of sEMG.

Fourth, in most dynamic US examinations, the anatomical structures are measured against the US transducer as a fixed point. To avoid the movement of the US transducer on the skin, we measured against an artificial reference with a reflective tape. Mohr et al. gave this approach an excellent reliability of ICC = 0.98 [41]. Nevertheless, it cannot be completely ruled out that skin movement affects the measurement. However, Langevin et al. [11] showed that almost no movement of the dermis was detectable during the shear motion of the TLF fascia layers. Therefore, we assume that skin motion was negligible for our results.

Finally, it is important to understand that PCA smoothing of the sEMG data, which was used in this study to avoid autocorrelation within the signal, does not allow conclusions to be drawn about neuromuscular control, although it cannot be considered a limitation in relation to the aim of this study. The approach of prewhitening the signals compared by CCR is strongly recommended [29]. 

The results of this study should be considered in light of the objective because, to the authors’ knowledge, this is the first study to investigate the relationships between TLF deformation and ESM activation in patients with aLBP and healthy control subjects. The results show that the myofascial changes caused by aLBP have a crucial influence on the sEMG behaviour of the ESM. We also hypothesize that a possible mechanism behind the observed effects may be adhesions between the fascial layers of the TLF and ESM epimysium, which alters muscle spindle function. Partanen et al. [55] recently presented a new method for identifying muscle spindle activity using EMG. It would be interesting to directly investigate the relationships between fascial deformation and muscle spindle activation using this method in a future study and confirm our hypothesis.

## 5. Conclusions

The results of this study suggest that associations between TLF deformation and ESM activity during a clinically relevant task can be revealed using CCR and GC comparisons of dynamic US imaging and sEMG data signals. Our results show that in aLBP, ESM activity is significantly affected by TLF, whereas this relationship is completely random in healthy individuals. We therefore hypothesize that adhesions between the fascial layers of the TLF and the epimysium of the ESM affect neuromotor control in aLBP, which could be triggered by altered muscle spindle function.

## Figures and Tables

**Figure 1 life-12-01735-f001:**
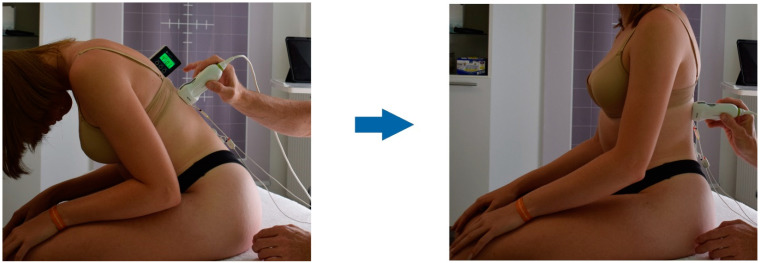
Trunk extension task. First, a 45° flexion position of the trunk is assumed; extension to the neutral position is performed using a computer-generated time announcement (8 s).

**Figure 2 life-12-01735-f002:**
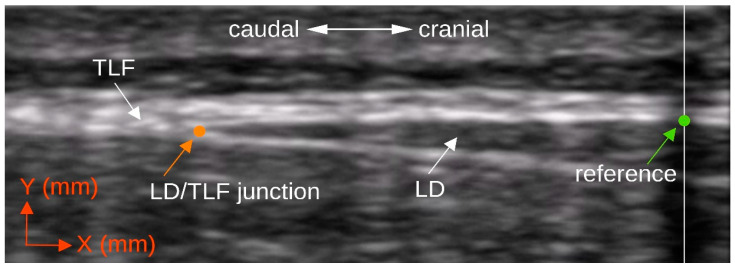
Single image measurement of *TLF* deformation. LD/TLF, junction, coordinate point *X*1|*Y*1 (orange dot) of the junction of LD to TLF; reference, coordinate point *X*2|*Y*2 (green dot) at the bottom of the TLF and centred on the US shadow; US, ultrasound; LD, latissimus dorsi muscle; TLF, thoracolumbar fascia.

**Figure 3 life-12-01735-f003:**
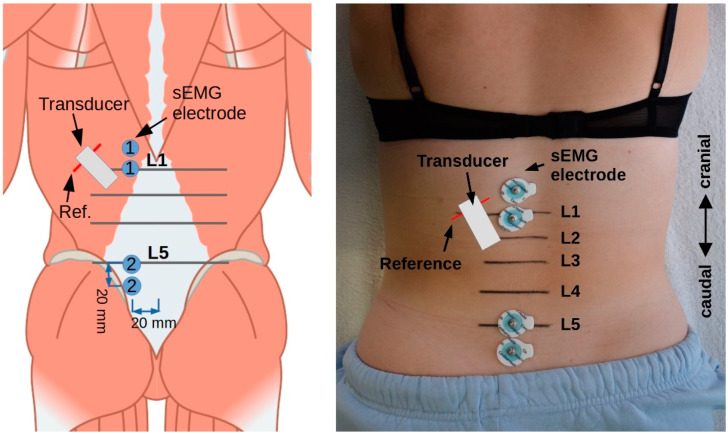
Measurement setup. Transducer, US transducer; Ref, artificial reference point; sEMG, surface electromyography. Picture adapted (https://creativecommons.org/licenses/by-sa/4.0/deed.en (accessed on 23 March 2022)).

**Figure 4 life-12-01735-f004:**
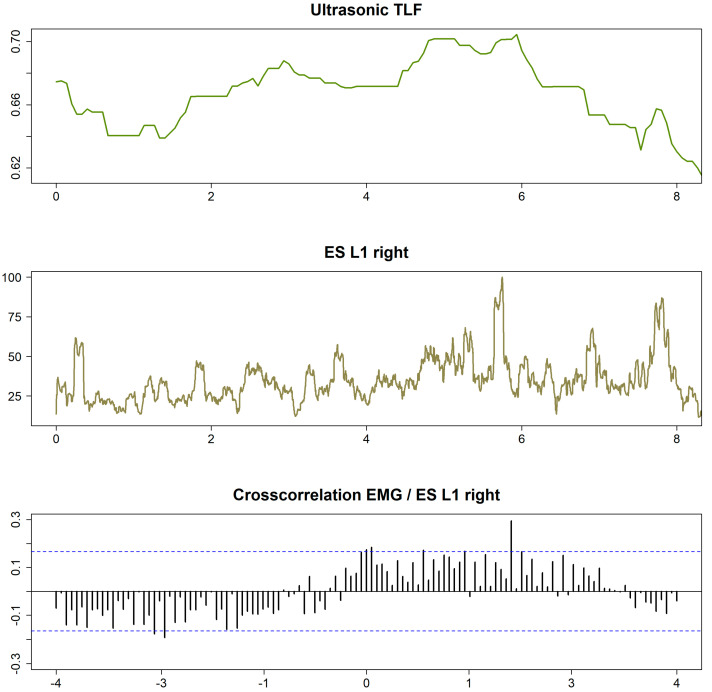
Cross-correlation analysis of the measurement series. Ultrasound TLF, deformation of TLF; ES L1 right, sEMG of right ES at L1 level; cross-correlation EMG / ES L1 right, cross-correlation function of the two time series (shows a significant time lag at 1.9 s in this example, indicating that the sEMG and deformation series are correlated). TLF, thoracolumbar fascia; EMG, electromyography; ES, erector spinae muscle.

**Figure 5 life-12-01735-f005:**
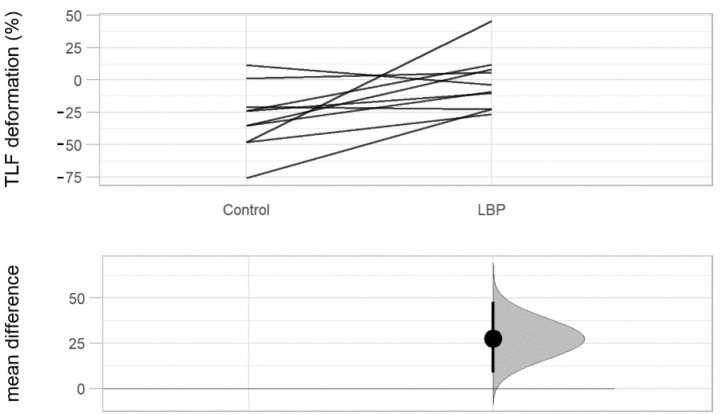
Estimation plot of TLF deformation. TLF, thoracolumbar fascia; LBP, acute low back pain group; Control, control group; significant at level *p* < 0.05.

**Figure 6 life-12-01735-f006:**
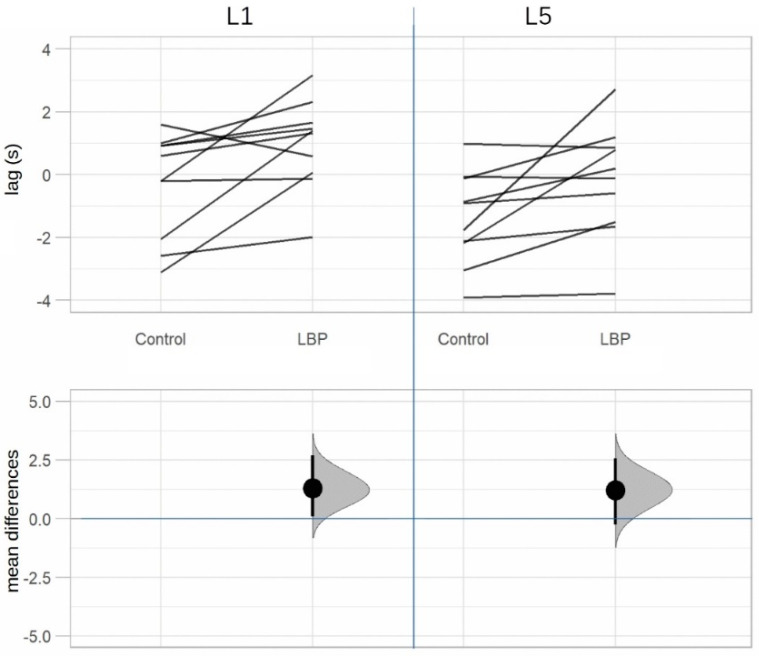
Estimation plot of time lags between the TLF deformation and the sEMG activity of the paraspinal muscles. TLF, thoracolumbar fascia; sEMG, surface electromyography; LBP, acute low back pain group; Control, control group; both significant at level *p* < 0.05.

**Figure 8 life-12-01735-f008:**
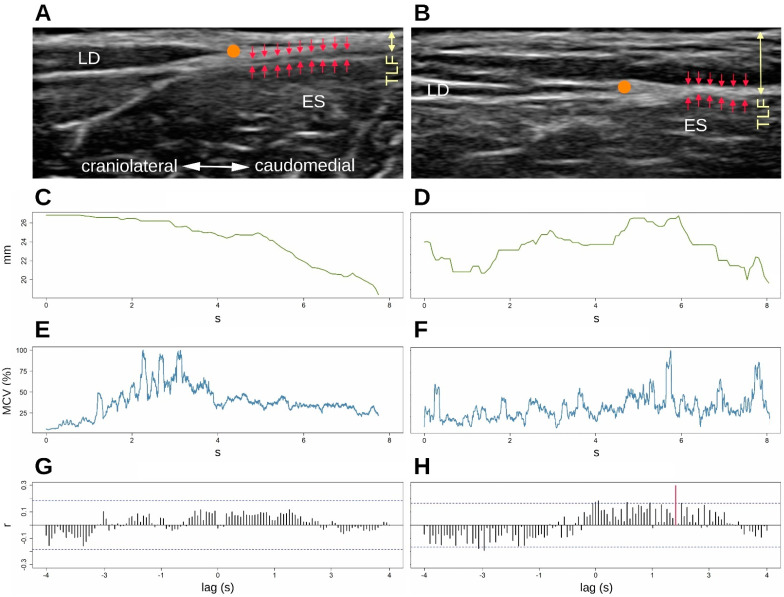
Differences of associations between patients with low back pain (**B**,**D**,**F**,**H**) and healthy control subjects (**A**,**C**,**E**,**G**). (**A**) A clear separation of the TLF and the epimysium of the ES by areolar connective tissue is shown (red arrows). (**B**) No clear separation is detectable (red arrows). In addition, the TLF appears distinctly thickened and disorganized (yellow arrows). (**C**) The diagram shows a gradual decrease in the deformation of the pre-extended thoracolumbar fascia with trunk extension. Surface EMG (**E**) shows no significant cross correlations (**G**), correlation coefficients all below 2/√n. (**D**) The thoracolumbar fascia behaves completely differently. It does not show a continuous behaviour but changes its deformation several times immediately in the course, causing a significant change in the sEMG activity (**F**) with a time shift of 1.9 s (**H**), red time delay. ES, erector spinae muscle; LD, latissimus dorsi muscle; SPI, serratus posterior inferior muscle; TLF, thoracolumbar fascia; sEMG, surface electromyography; orange dot shows the LD/TLF junction.

**Figure 9 life-12-01735-f009:**
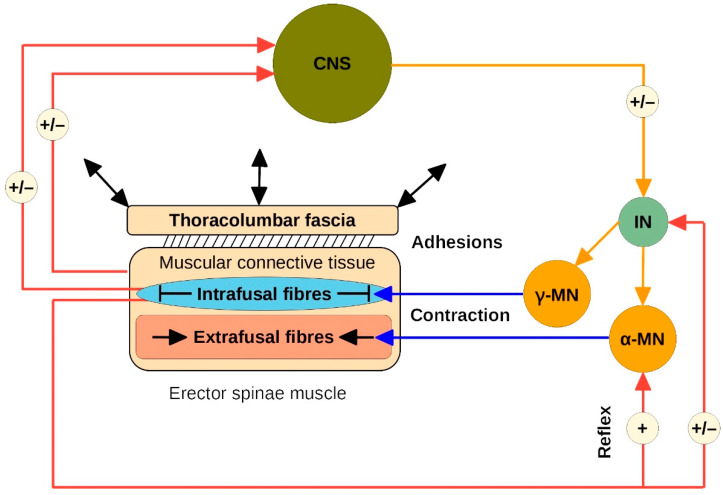
Pathological fascial triggered alteration of muscle spindle function.

**Table 1 life-12-01735-t001:** Baseline characteristics.

Baseline Characteristics	LBP Group (*n* = 10) Mean ± SD	Controls (*n* = 10) Mean ± SD	*p* Value
Sex (men/women)	4/6	4/6	
Age (years)	43.6 ± 15.9	39.0 ± 15.0	0.38 ^a^
95% CI	32.2–54.9	28.2–49.7	
Min–Max	18.5–59.6	18.–58.0	
Height (m)	1.72 ± 0.08	1.67 ± 0.12	0.30
95% CI	1.67–1.78	1.59–1.76	
Min–Max	1.61–1.83	1.51–1.83	
Weight (kg)	74.5 ± 12.9	66.0 ± 15.6	0.16
95% CI	66.3–84.7	54.9–77.2	
Min–Max	57.0–98.0	48.0–99.0	
BMI (kg/m^2^)	25.6 ± 4.5	23.5 ± 4.6	0.24 ^a^
95% CI	22.3–28.8	20.3–26.7	
Min–Max	19.6–33.5	19.5–34.3	
ODQ-D (0–100)	49.4 ± 16.8		
95% CI	37.4–61.4		
Min–Max	32.0–78.0		
VAS (0–10)	5.3 ± 2.5		
95% CI	3.5–7.0		
Min–Max	2.6–10.0		
Pain duration (days)	9.1 ± 3.7		
95% CI	6.4–11.8		
Min–Max	3.0–14.0		

Normally distributed data are tested with Student’s *t*-test. ^a^, not normally distributed data, tested with Mann–Whitney U test. SD, standard deviation; n, number; 95% CI, 95% confidence interval; LBP, low back pain; ODQ-D, Oswestry disability questionnaire in the German version; VAS, Visual analogue scale.

**Table 2 life-12-01735-t002:** Changes between acute low back pain patients and healthy controls.

			Student’s *t*-Test		
Outcome	Mean ± SD	95% CI	t	*p*	d
TLF deformation (%)	27.5 ± 31.1	5.3–49.8	2.80	**0.01 ***	0.88
L1 time lag (s)	1.30 ± 1.53	0.21–2.39	2.69	**0.02 *** (0.04 **)	0.85
L5 time lag (s)	1.21 ± 1.49	0.15–2.28	2.57	**0.03 *** (0.04 **)	0.81
L1 TLF/sEMG (%)	30.6 ± 44.6	−1.2–62.4	2.17	**0.03 *** (0.03 **)	0.69
L5 TLF/sEMG (%)	34.6 ± 40.6	5.5–63.6	2.69	**0.01 *** (0.02 **)	0.85
L1 sEMG/TLF (%)	2.8 ± 45.8	−30.0–35.5	0.19	0.85	0.06
L5 sEMG/TLF (%)	5.6 ± 43.3	−25.3–36.6	0.41	0.69	0.13

Values represent the differences between matched aLBP patients and healthy subjects. SD, standard deviation; n, number; 95% CI, 95% confidence interval; d, Cohen’s d; TLF, thoracolumbar fascia; TLF/sEMG, Granger causality that the deformation of the thoracolumbar fascia causes the sEMG; sEMG/TLF, Granger causality that the sEMG causes the deformation of the thoracolumbar fascia; aLBP, acute low back pain; TLF, thoracolumbar fascia; sEMG surface electromyography; significant at the level * *p* < 0.05; ** Bonferroni–Holm adjusted.

**Table 3 life-12-01735-t003:** Granger causality.

		Granger Prediction			
		L1		L5	
Direction	Group	True	False	True	False
TLF/sEMG	LBP	82.3%	17.7%	92.5%	7.5%
	Control	51.7%	48.3%	58.0%	42.0%
sEMG/TLF	LBP	61.0%	39.0%	52.3%	47.7%
	Control	58.2%	41.8%	46.7%	53.3%

LBP, low back pain group; Control, control group; TLF/sEMG, Granger causality that the deformation of the thoracolumbar fascia causes the sEMG; sEMG/TLF, Granger causality that the sEMG causes the deformation of the TLF; TLF, thoracolumbar fascia; sEMG, surface electromyography.

## Data Availability

Data can be made available by the author upon request.

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
