# Peer review of "Associations between Deformation of the Thoracolumbar Fascia and Activation of the Erector Spinae and Multifidus Muscle in Patients with Acute Low Back Pain and Healthy Controls: A Matched Pair Case-Control Study"

_life, 2022, doi:10.3390/life12111735_

Round 1

Reviewer 1 Report

I am glad to review the manuscript ‘Associations between Deformation of the Thoracolumbar 2 Fascia and Activation of the Erector Spinae Muscle in Patients 3 with Acute Low Back Pain and Healthy Controls. A Matched 4 Pair Case-Control Study’.

 In the abstract they coıncluded that in aLBP, ES activity is significantly affected by TLF. Conclusion needs to be clear how TFL affected ES activity.

Introduction: Patients with nonspecific LBP not only visit orthopedists but also primary care physisicans, physiatrists and neurosurgeons.

Patients with higher BMI and suffering from LBP, particularly women, had more fatty infiltration in their paraspinal muscles at upper lumbar levels, more severe IVDD and Modic changes at lower lumbar levels (ref: Obesity could be associated with poor paraspinal muscle quality at upper lumbar levels and degenerated spine at lower lumbar levels: Is this a domino effect? Özcan-EkÅŸi EE, Turgut VU, KüçüksüleymanoÄŸlu D, EkÅŸi MÅž. J Clin Neurosci. 2021 Dec;94:120-127. doi: 10.1016/j.jocn.2021.10.005. Epub 2021 Oct 20.PMID: 34863425)

Line 62. This information should be added.  It has been shown that patients with fatty multifidus were 4 times more likely to have chr LBP and psoas counteracts to stabilize the LBP ( doi: 10.1080/02688697.2020.1783434. Epub 2020 Jun 24.Emel Ece Özcan-EkÅŸi Murat Åžakir EkÅŸiVeli Umut TurgutÇaÄŸrı CanbolatM Necmettin Pamir. Reciprocal relationship between multifidus and psoas at L4-L5 level in women with low back pain. Br J Neurosurg. 2021 Apr;35(2):220-228.PMID: 32576034)

 Methods:

Why did the authors not evaluate multifidus?

Exclusion criteria: Did the authors exclude those with trauma, scoliosis, neurologic/psychiatric /endocrine disorders? If yes, they should mention.

Did the authors have intra- and inter-rater reliability for evlaauting TLF deformation using ultrasound?

It has been reproted that thoracic fibers of the erector spinae accounts for appr. 80% of the extensor moment of the upper lumbar spine. Thoracic fibers of the eretor spinae accounts for the 50% of the extension at the lower lumbar spine. (ref: N Bogduk 1J E MacintoshM J Pearcy. A universal model of the lumbar back muscles in the upright position. Spine (Phila Pa 1976)

. 1992 Aug;17(8):897-913. PMID: 1523493). Why did the authors meausred activity at lower lumbar levels for the erector spinae?

 Limitation part should include why the authors did not evaluate the multifidus and psoas.

Conclusion: They coud not use the  term ‘paraspinal muscles’, since they evlauated only the erector spinae. They should correct the conclusion part.

Author Response

We were very impressed with the thorough review and the many valuable recommendations and improvements we received from you. We are convinced that this will significantly improve our article quality. We have revised our article in this respect, following step by step the recommendations of your review. Below we provide a point-by-point commentary your comments.

  1. In the abstract they coıncluded that in aLBP, ES activity is significantly affected by TLF. Conclusion needs to be clear how TFL affected ES activity.
    In accordance with the general recommendations[1] for the sections with scientific conclusions, we have summarized our thoughts and conclusions. A detailed model of our considerations is presented in the discussion section. Lines 344-361; lines 362-378, lines 379-399.
  2. Introduction: Patients with nonspecific LBP not only visit orthopedists but also primary care physisicans, physiatrists and neurosurgeons.
    We agree. Our statement refers to the references[2] and focuses on the epidemiological data. Therefore, primary care providers with initial access are mainly orthopedists (particularly in Europe and the USA).
  3. Patients with higher BMI and suffering from LBP, particularly women, had more fatty infiltration in their paraspinal muscles at upper lumbar levels, more severe IVDD and Modic changes at lower lumbar levels (ref: Obesity could be associated with poor paraspinal muscle quality at upper lumbar levels and degenerated spine at lower lumbar levels: Is this a domino effect? Özcan-EkÅŸi EE, Turgut VU, KüçüksüleymanoÄŸlu D, EkÅŸi MÅž. J Clin Neurosci. 2021 Dec;94:120-127. doi: 10.1016/j.jocn.2021.10.005. Epub 2021 Oct 20.PMID: 34863425)
    This is an interesting aspect in LBP genesis. However, it was not our focus in this study, so we used a matched-pairs design and tested baseline characteristics. There were no significant differences between our case and control groups in age, height, weight, or BMI (in your reference, higher BMI was associated with intervertebral disc degeneration (IVDD)).
  4. Line 62. This information should be added. It has been shown that patients with fatty multifidus were 4 times more likely to have chr LBP and psoas counteracts to stabilize the LBP ( doi: 10.1080/02688697.2020.1783434. Epub 2020 Jun 24.Emel Ece Özcan-EkÅŸi , Murat Åžakir EkÅŸi, Veli Umut Turgut, ÇaÄŸrı Canbolat, M Necmettin Pamir. Reciprocal relationship between multifidus and psoas at L4-L5 level in women with low back pain. Br J Neurosurg. 2021 Apr;35(2):220-228.PMID: 32576034)
         We agree that there are a variety of risk factors for the development of LBP. We focused on acute LBP because current research suggests that this is the highest risk factor. For our statement, we reviewed one high-quality umbrella review and one systematic review with a total of 152 cohort or randomized control trials.
  5. Why did the authors not evaluate multifidus?
    We examined the longissimus muscle (at L1 level) and the multifidus muscle (at L5 level) according to the SENIAM protocol (lines 219, 220). To clarify this fact, we have added "multifidus" to the title and changed the abbreviation ES to ESM (erector spinae and multifidus muscle).
  6. Exclusion criteria: Did the authors exclude those with trauma, scoliosis, neurologic/psychiatric /endocrine disorders? If yes, they should mention.
    We extended our exclusion criteria to include acute trauma, neurological or psychiatric disorders that we knew no subject participating in the study had. We did not check our subjects for endocrine disorders, but we did check the medications they were taking, and there were no medications for endocrine disorders. We therefore added this criterion. Lines 145-148.
  7. Did the authors have intra- and inter-rater reliability for evlaauting TLF deformation using ultrasound?
    We have added a reference in this regard. Lines 175-177.
  8. It has been reproted that thoracic fibers of the erector spinae accounts for appr. 80% of the extensor moment of the upper lumbar spine. Thoracic fibers of the eretor spinae accounts for the 50% of the extension at the lower lumbar spine. (ref: N Bogduk 1, J E Macintosh, M J Pearcy. A universal model of the lumbar back muscles in the upright position. Spine (Phila Pa 1976) . 1992 Aug;17(8):897-913. PMID: 1523493). Why did the authors meausred activity at lower lumbar levels for the erector spinae?
    In the aforementioned article from 1992, moments of force were calculated on an artificial lumbar vertebra model based on anatomical data from 1986. More recent research[3] suggests a different view of the back, in particular the erector spinae and the multifidus muscle. Our aim was to investigate the influence of the TLF on the activity of the ES and multifidus. Therefore, we measured the deformation of the TLF and the muscle activation under the TLF in the lumbar region.
  9. Limitation part should include why the authors did not evaluate the multifidus and psoas.
    The psoas muscle was not our focus. The multifidus muscle was measured. Our goal was to investigate the influence of TLF on the activity of the ES and multifidus. Therefore, we omitted the psoas muscle because it is not measurable with surface EMG and is not directly connected to the posterior layer of the TLF.

Conclusion: They coud not use the term ‘paraspinal muscles’, since they evlauated only the erector spinae. They should correct the conclusion part.
We have changed this expression to EMS. Line 468.

[1] e.g. Sacred Heart University: Organizing Academic Research Papers; https://library.sacredheart.edu/c.php?g=29803&p=185935

[2] Levy, V.J.; Holt, C.T.; Haskins, A.E. Osteopathic Manipulative Medicine Consultations for Hospitalized Patients. J Am Osteopath Assoc 2019, 119, 299–306, doi:10.7556/jaoa.2019.051.

van Tulder, M.; Becker, A.; Bekkering, T.; Breen, A.; Gil del Real, M.T.; Hutchinson, A.; Koes, B.; Laerum, E.; Malmivaara, A. European Guidelines for the Management of Acute Nonspecific Low Back Pain in Primary Care. Eur Spine J 2006, 15, s169–s191, doi:10.1007/s00586-006-1071-2.

[3] Creze M, Soubeyrand M, Gagey O (2019) The paraspinal muscle-tendon system: Ist paradoxical anatomy. PLoS ONE 14(4): e0214812. https://doi.org/10.1371/journal.pone.0214812

Reviewer 2 Report

Associations between Deformation of the Thoracolumbar Fascia and Activation of the Erector Spinae Muscle in Patients with Acute Low Back Pain and Healthy Controls. A Matched Pair Case-Control Study

Andreas Brandl 1,2,3,*, Christoph Egner 2 , Rüdiger Reer 1 , Tobias Schmidt 3,4 , Robert Schleip 2,5

JCM

This is a nice study performed with well chosen methodology to demonstrate TLF deformation as a potential cause of acute LBP. It is innovative and raises relevant insights into the etiology of acute LBP. However with respect to their explanation authors present an important figure with seperated explanantion that confuses me.

Abstract

L33 Granger causality: Is it rather Granger predictive causalty?

1.     Introduction

The introduction is quite long, to my opinion. There are topics discussed that can be moved to the discussion section. The explanation or background leading to the study questions can be managed more compactly.

2. Materials and Methods

2.2 Setting and participants

L132 Did authors consider only including patients with normal BMI? Why or why not?

L137 (c) a minimum 137 score of 3 on the visual analogue scale (VAS). I suggest to add pain to scale: the visual analogue pain scale 

L185-191 Please give more explanation with respect tot the coordination points X0-1, Y0-1 etc. Not all readers will understand.

2.5 Statistics

Stutent’s T-test should be Student’s T-test?

L260 Statis-tical: delete –

With respect of the use of sEMG it is stated  that surface EMG of ES and LM are no adequate representation of LM and ES activity measured by iEMG because of moderate/high cross-talk and co-contractions (Hofste A, et al, Spine (Phila Pa 1976). 2020 Oct 15;45(20):E1319-E1325. doi: 10.1097/BRS.0000000000003624). How do authors comment on that in relation to the use of sEMG to assess ES in their own study?

2.     Results

Table 1: Gender is not the correct word to use in this context. I suggest to replace it by sex.

L284 (Granger prediction) and GC: please use consequently the same terms throughout the manuscript in stead of Granger causalty and Granger prediction.

L310-312. In addition, this is the first study to use CCR functions and GC, which are commonly used in signal processing analysis in psychophysiology and neuroscience [27]. Whittaker et al. [25] used a similar methodology to compare abdominal muscle thickness and EMG activity.

This is confusing, because later the authors state: To the authors' knowledge, this was the first study to use CCR functions and GC to reveal the interactions between TLF and ES in pathological and healthy subjects. So being the first study accounts for the context of this study. I suggest the authors to rephrase this more accurately.

L329-332 Wong et al. [36] observed a difference in TLF deformation of 3.8 mm after myofascial release treatment. Therefore, our result of only a nonsignificant difference of 0.23 mm between groups was surprising.

Can authors explain if there were differences between Wongs methodology and the methodology of the autors’?

L354-355 B, no clear separation is detectable (red arrows). In addition, the TLF appears distinctly thickened and disorganized (turquoise arrows).

-          The colors are hardly to distinguish from eachother; please change colors

-          The TLF appears distinctly thickened and disorganized: does this account for healthy subjects? Because this sentence refers to B. I’m confused about this. Please explain.

-          Is TLF thickening characteristic for healthy subjects?

L370-371 Adhesions of the TLF to the epimysium of the ES reduce the elasticity and adaptability of the intramuscular connective tissue to which the spindles are connected.

It seems that this accounts for the figures at the right side, while these are labeled healthy controls. This is confusing. Please explain.

5. Conclusions

In the explaining theory, authors state that the TLF deformation must be the underlying problem for dysfunctioning muscle contraction. Can authors give an explanantion from clinical etiological perspective why the causal problem should lie in the TLF? They refer to TLF adhesions. Why should adhesions occur as first etiological cause?

Author Response

We were very impressed with the thorough review and the many valuable recommendations and improvements we received from you. We are convinced that this will significantly improve our article quality. We have revised our article in this respect, following step by step the recommendations of your review. Below we provide a point-by-point commentary your comments.

  1. L33 Granger causality: Is it rather Granger predictive causalty?
    Granger causality is the commonly used term[1]. In other sections of our paper, we have changed the term "Granger prediction" to "Granger causality" to unify our language.
  2. The introduction is quite long, to my opinion. There are topics discussed that can be moved to the discussion section. The explanation or background leading to the study questions can be managed more compactly.
    We thoroughly planned our study based on current knowledge according to STROBE Statement[2]. Therefore, we referred to demographic data on the prevalence of nonspecific low back pain, the background for suspected associations between TLF and muscle activation of the erector spinae and multifidus muscles, and our methodological considerations. Therefore, according to the STROBE checklist, item 2 "Explain the scientific background and rationale for the investigation being reported" and 3 “State specific objectives, including any prespecified hypotheses” we included this information in the background section rather than the discussion section.
         Due to the novel approach, more background information was required than in a study based on already known methodological considerations. Nevertheless, our introduction section of 678 words (excluding references and objectives) is slightly above the mean of all scientific papers (553 words) and well within the 75th percentile of all scientific papers (762 words)[3].
  3. L132 Did authors consider only including patients with normal BMI? Why or why not?
    We included participants with BMI between 18.5 and 34.9 (normal to grade 1 obesity) and matched controls for BMI to reflect LBP patients with higher BMI, which is often the case in practice. We highlighted this more clearly in our inclusion criteria (line 141) and in the table (minimum and maximum values) of baseline characteristics (line 290).
  4. L137 (c) a minimum 137 score of 3 on the visual analogue scale (VAS). I suggest to add pain to scale: the visual analogue pain scale
    We added " for assessment of pain intensity" to VAS because visual analogue scale is the most commonly used term (29,644 search results for "visual analogue scale" on MEDLINE vs. 565 for "visual analogue pain scale"). Line 140.
  1. L185-191 Please give more explanation with respect tot the coordination points X0-1, Y0-1 etc. Not all readers will understand.
    We have clarified this point and explained the coordinate points. We have also added a reference to the points in Figure 2. Lines 191-195.
  2. Stutent’s T-test should be Student’s T-test?
    We have corrected this spelling error. Lines 263, 270.
  3. L260 Statis-tical: delete –
    We deleted the hyphen.
  4. With respect of the use of sEMG it is stated that surface EMG of ES and LM are no adequate representation of LM and ES activity measured by iEMG because of moderate/high cross-talk and co-contractions (Hofste A, et al, Spine (Phila Pa 1976). 2020 Oct 15;45(20):E1319-E1325. doi: 10.1097/BRS.0000000000003624). How do authors comment on that in relation to the use of sEMG to assess ES in their own study?
         We also considered the work of Hofste et al. to evaluate the reliability of the sEMG. A close look at the paper shows a correlation of r = 0.9 between the iEMG and the sEMG of the erector spinae muscle and an r = 0.8 for the multifidus muscle during a biofeedback task. Other correlations were elicited by coactivation tasks that did not follow the SENIAM protocol. However, we have added a comment on this uncertainty in our limitations section. Lines 436-442
  5. Table 1: Gender is not the correct word to use in this context. I suggest to replace it by sex.
    We changed the word to sex. Line 290.
  6. L284 (Granger prediction) and GC: please use consequently the same terms throughout the manuscript in stead of Granger causalty and Granger prediction.
         Granger causality is the commonly used term[4]. In other sections of our paper, we have changed the term "Granger prediction" to "Granger causality" to unify our language.
  7. L310-312. In addition, this is the first study to use CCR functions and GC, which are commonly used in signal processing analysis in psychophysiology and neuroscience [27]. Whittaker et al. [25] used a similar methodology to compare abdominal muscle thickness and EMG activity. This is confusing, because later the authors state: To the authors' knowledge, this was the first study to use CCR functions and GC to reveal the interactions between TLF and ES in pathological and healthy subjects. So being the first study accounts for the context of this study. I suggest the authors to rephrase this more accurately.
    We have worded this sentence with additional information to clarify that the new approach is to apply CCR to myofascial lumbar tissues (whereas it is already being applied to other tissues and specifically in neuroscience). Lines 332, 333.
  8. L329-332 Wong et al. [36] observed a difference in TLF deformation of 3.8 mm after myofascial release treatment. Therefore, our result of only a nonsignificant difference of 0.23 mm between groups was surprising. Can authors explain if there were differences between Wongs methodology and the methodology of the autors’?
         We explained the results of Wong et al. in more detail and also adjusted our deformation measurement according to the methodology of Wong et al. Therefore, we demonstrated a significant difference between LBP patients and healthy control subjects. We have modified our discussion in this regard. Lines 295, 305, 353-361.
  9. L354-355 B, no clear separation is detectable (red arrows). In addition, the TLF appears distinctly thickened and disorganized (turquoise arrows).
    1. The colors are hardly to distinguish from eachother; please change colors
      We have changed the colors to yellow, which is easier to see. We have also enlarged the figure slightly for better readability and changed the color for images E and F to distinguish between the TLF deformation curve and the sEMG.
    2. The TLF appears distinctly thickened and disorganized: does this account for healthy subjects? Because this sentence refers to B. I’m confused about this. Please explain.
      This was incorrectly explained. Images B, D, F, H show the subject with low back pain and A, C, E, G show the healthy subject. We have corrected this. Line 407.
    3. Is TLF thickening characteristic for healthy subjects?
      See answer 13b.
  10. L370-371 Adhesions of the TLF to the epimysium of the ES reduce the elasticity and adaptability of the intramuscular connective tissue to which the spindles are connected. It seems that this accounts for the figures at the right side, while these are labeled healthy controls. This is confusing. Please explain.
    This was incorrectly explained. Images B, D, F, H show the subject with low back pain and A, C, E, G show the healthy subject. We have corrected this. Line 408.
  11. In the explaining theory, authors state that the TLF deformation must be the underlying problem for dysfunctioning muscle contraction. Can authors give an explanantion from clinical etiological perspective why the causal problem should lie in the TLF? They refer to TLF adhesions. Why should adhesions occur as first etiological cause?
    We have referred to this in the introduction (lines 69-75) to provide background information for our hypothesis (lines 101-118). We have also elaborated that the underlying mechanisms are not fully understood to date, that is, why we are testing in our study whether fascia influence muscle activity or vice versa[5].

[1] Granger, C. W. J. (1969). "Investigating Causal Relations by Econometric Models and Cross-spectral Methods". Econometrica. 37 (3): 424–438. doi:10.2307/1912791. JSTOR 1912791.
See also: https://en.wikipedia.org/wiki/ Granger_causality

[2] von Elm, E.; Altman, D.G.; Egger, M.; Pocock, S.J.; Gøtzsche, P.C.; Vandenbroucke, J.P. The Strengthening the Reporting of Observational Studies in Epidemiology (STROBE) Statement: Guidelines for Reporting Observational Studies. Ann Intern Med 2007, 147, 573–577, doi:10.7326/0003-4819-147-8-200710160-00010.

[3] Comeau DC, Wei CH, Islamaj DoÄŸan R, and Lu Z. PMC text mining subset in BioC: about 3 million full text articles and growing, Bioinformatics, btz070, 2019.

[4] Granger, C. W. J. (1969). "Investigating Causal Relations by Econometric Models and Cross-spectral Methods". Econometrica. 37 (3): 424–438. doi:10.2307/1912791. JSTOR 1912791.
See also: https://en.wikipedia.org/wiki/ Granger_causality

[5] Langevin, H.M.; Fox, J.R.; Koptiuch, C.; Badger, G.J.; Greenan- Naumann, A.C.; Bouffard, N.A.; Konofagou, E.E.; Lee, W.-N.; Triano, J.J.; Henry, S.M. Reduced Thoracolumbar Fascia Shear Strain in Human Chronic Low Back Pain. BMC Musculoskelet Disord 2011, 12, 203, doi:10.1186/1471-2474-12-203.

Wilke, J.; Schleip, R.; Klingler, W.; Stecco, C. The Lumbodorsal Fascia as a Potential Source of Low Back Pain: A Narrative Review. BioMed Research International 2017, 2017, 1–6, doi:10.1155/2017/5349620.

Stecco, A.; Gesi, M.; Stecco, C.; Stern, R. Fascial Components of the Myofascial Pain Syndrome. Curr Pain Headache Rep 2013, 17, 352, doi:10.1007/s11916-013-0352-9.

Reviewer 3 Report

Dear authors.

Congratulations for the idea and the work presented it is quiet interesting.

Attached can find the comments, suggestions and questions revealed when read the document.

Regards.

Author Response

We were very impressed with the thorough review and the many valuable recommendations and improvements we received from you. We are convinced that this will significantly improve our article quality. We have revised our article in this respect, following step by step the recommendations of your review. Below we provide a point-by-point commentary your comments.

  1. Page 2, lines 90-91: This affirmation is poor taken into acount the sampling of this work (n=7). By other side there are other authors which reached correlation. Review this literature as example:
    Ferreira, P. H. et al. Discriminative and reliability analyses of ultrasound measurement of abdominal muscles recruitment. Man. Ther. 16, 463–469 (2011).
         McMeeken, J. M., Beith, I. D., Newham, D. J., Milligan, P. & Critchley, D. J. The relationship between EMG and change in thickness of transversus abdominis. Clin. Biomech. 19, 337–342 (2004).
         Kim, C. Y. et al. Comparison between muscle activation measured by electromyography and muscle thickness measured using ultrasonography for effective muscle assessment. J. Electromyogr. Kinesiol. 24, 614–620 (2014).

       We have taken your reference suggestion into account and have reworded our sentence. Lines 87, 92.
  2. Page 4, lines 179-181: Include a surface photo with the model with the figure 3 for better understanding
    We have added a surface photo. Line 243.
  3. Page 4, lines 184-185: 1. You are agreeing the insertion of LD in the underside of the posterior layer of the TLF. But you have to:
    1. Detail an anatomical reference whichs describes this insertion.
      We have added the references to the anatomical description. Line 192.
    2. Support this ultrasound approach with a previous which has demonstrated ICC validity enough.
      We have added the reference. Lines 175-177.
    3. Considering this statement, the distance measured was between points green and orange in Figure 2.
      If this statement were true, the point green is the reference I is not sure could be the reference due to the skin can be moved over the subcutaneous layer. This could be a bias, so:
      - Has been considered this thoughts?
         - How did you solve this consideration?
      In ultrasound examinations, the transducer is the fixed point at which relative measurements are taken. In dynamic ultrasound imaging, it is therefore of paramount importance to prevent unintentional movement of the transducer - this represents the greatest risk of systematic bias. Mohr et al.[1] therefore investigated the reflective tape method (which we used) in terms of its reliability with a well-established three-dimensional motion analysis system and found excellent correlations (ICC = 0.98).
           However, you are correct, it cannot be completely ruled out that skin movements influence the measurement. We have outlined this and our thoughts on it in the "Limitations" section. Lines 443-450.
  4. Page 5, lines 190-191: Detail in this foot image whiches are green and orange points in the formula.
    We have added the description. Lines 191, 192, 200.
  5. Page 6, lines 222-226: More details about how the signals where sinchronized are needed.
    1. Which program was used?
      We have documented the entire procedure in more than this section. In 2.4.3, Data synchronization, only the technical procedure of signal synchronization is described. We have described the procedures for dynamic ultrasound analysis in detail in Section 2.4.1 and for surface electromyography in Section 2.4.2. The software analysis of the cross-correlation between the two signals is described in detail in Section 2.4.4. We have additionally included the detailed information about the software we used. For the ultrasound analysis: lines 207-209; for the EMG analysis: lines 234-235.
    2. Include supplementary material with video how it works with detailed information about the two signals (EMG and US) working.
      We have added three additional videos to demonstrate the technique of video analysis and provide more background information on dynamic ultrasound results. Lines 474-478.

[1] Mohr, L.; Vogt, L.; Wilke, J. Use of Reflective Tape to Detect Ultrasound Transducer Movement: A Validation Study. Life 2021, 11, 104, doi:10.3390/life11020104.

Round 2

Reviewer 3 Report

Dear authors.

The document has been highly enhanced but you forgot to answer the first commentary ant the begining of the first round and now I come back to detail here which I consider very important, due to is a base of the conceptual framework of this research (lines 91-93) where is written:

"Whittaker et al. [26] could not find any correlation even  between EMG data of abdominal muscles and their thickness measured with US during an abdominal drawing-in maneuver or an active straight leg raise test. "

My commentary was:

This affirmation is poor taken into account the sampling of this work (n=7). By other side there are other authors which reached correlation. Review this literature as example:

-Ferreira, P. H. et al. Discriminative and reliability analyses of ultrasound measurement of abdominal muscles recruitment. Man. Ther. 16, 463–469 (2011).

-McMeeken, J. M., Beith, I. D., Newham, D. J., Milligan, P. & Critchley, D. J. The relationship between EMG and change in thickness of transversus abdominis. Clin. Biomech. 19, 337–342 (2004).

-Kim, C. Y. et al. Comparison between muscle activation measured by electromyography and muscle thickness measured using ultrasonography for effective muscle assessment. J. Electromyogr. Kinesiol. 24, 614–620 (2014).

Considering the results of this papers and knowing there are more, the statement detailed in lines 91-93 would be not correct. Reconsider this point and solve it.

Author Response

Thank you again for the detailed review. Below we provide a point-by-point commentary on your comment.

Comment:

  1. The document has been highly enhanced but you forgot to answer the first commentary ant the begining of the first round and now I come back to detail here which I consider very important, due to is a base of the conceptual framework of this research (lines 91-93) where is written:
    "Whittaker et al. [26] could not find any correlation even  between EMG data of abdominal muscles and their thickness measured with US during an abdominal drawing-in maneuver or an active straight leg raise test. "

    My commentary was:
    This affirmation is poor taken into account the sampling of this work (n=7). By other side there are other authors which reached correlation. Review this literature as example:

    -Ferreira, P. H. et al. Discriminative and reliability analyses of ultrasound measurement of abdominal muscles recruitment. Man. Ther. 16, 463–469 (2011).

    -McMeeken, J. M., Beith, I. D., Newham, D. J., Milligan, P. & Critchley, D. J. The relationship between EMG and change in thickness of transversus abdominis. Clin. Biomech. 19, 337–342 (2004).

    -Kim, C. Y. et al. Comparison between muscle activation measured by electromyography and muscle thickness measured using ultrasonography for effective muscle assessment. J. Electromyogr. Kinesiol. 24, 614–620 (2014).

    Considering the results of this papers and knowing there are more, the statement detailed in lines 91-93 would be not correct. Reconsider this point and solve it.

    We have considered your recommendation in lines 87 and 88 and also added the references you suggested. We cited the Whittaker et al. paper because they used a similar cross-correlation approach. However, we agree with you that this is more of a lower quality experiment than the work you suggested. Therefore, we have deleted the explicit citation of Whittaker's work. Lines 91-92.

Round 3

Reviewer 3 Report

Congratulations, it is a great work.